# DLP Fabrication of Zirconia Scaffolds Coated with HA/β-TCP Layer: Role of Scaffold Architecture on Mechanical and Biological Properties

**DOI:** 10.3390/jfb13030148

**Published:** 2022-09-12

**Authors:** Bartolomeo Coppola, Laura Montanaro, Paola Palmero

**Affiliations:** INSTM R.U. Lince Laboratory, Department of Applied Science and Technology, Politecnico di Torino, Corso Duca Degli Abruzzi 24, 10129 Torino, Italy

**Keywords:** scaffolds, zirconia, HA/β-TCP, DLP, 3D printing, SBF, mechanical properties

## Abstract

In order to merge high-mechanical properties and suitable bioactivity in a single scaffold, zirconia porous structures are here coated with a hydroxyapatite layer. The digital light processing (DLP) technique is used to fabricate two types of scaffolds: simple lattice structures, with different sizes between struts (750, 900 and 1050 µm), and more complex trabecular ones, these latter designed to better mimic the bone structure. Mechanical tests performed on samples sintered at 1400 °C provided a linear trend with a decrease in the compressive strength by increasing the porosity amount, achieving compressive strengths ranging between 128–177 MPa for lattice scaffolds and 34 MPa for trabecular ones. Scaffolds were successfully coated by dipping the sintered samples in a hydroxyapatite (HA) alcoholic suspension, after optimizing the HA solid loading at 20 wt%. After calcination at 1300 °C, the coating layer, composed of a mixture of HA and β-TCP (β-TriCalcium Phospate) adhered well to the zirconia substrate. The coated samples showed a proper bioactivity, well pronounced after 14 days of immersion into simulated body fluid (SBF), with a more homogeneous apatite layer formation into the trabecular samples compared to the lattice ones.

## 1. Introduction

Bioceramics are a class of technical ceramics characterized by low toxicity and high biocompatibility, making their use in medical applications wider and wider.

Bioceramics can be classified as inert and bioactive ceramics. The former, which include alumina, zirconia and their composites, are characterized by high mechanical properties, which make them suitable prosthetic materials for orthopedics and dentistry [1]. However, they are unable to form chemical bonds with natural bone and to actively integrate into the human body [2]. In this field, zirconia ceramics show outstanding flexural strength and high fracture toughness thanks to the tetragonal to monoclinic phase transformation [3]. On the other hand, bioactive ceramics, such as hydroxyapatite (HA) and—more generally—calcium phosphates [2], are able to elicit a biological response from the surrounding living system, which results in the formation of a bond between the tissue and the material. Such a unique property makes them excellent candidates for bone regeneration, also due to the chemical affinity of hydroxyapatite to the mineral part of bone. Indeed, the stoichiometric HA (Ca:P molar ratio of 1.67) has the following chemical formula: Ca_10_(PO_4_)_6_(OH)_2_ [4]. Hydroxyapatite has numerous applications, in the chemical [5,6], optical [7] and electronics [8,9] industries, but its main use is in medicine [10]. In particular, HA is widely used in the field of bone restoration and calcium phosphate ceramics are usually shaped as macro-porous materials, named scaffolds, with the porosity amount and size designed to achieve the correct vascularization and cells interaction, as the basis for new bone formation. However, these bioactive ceramics are characterized by low strength and fracture toughness, limiting their application to non-bearing parts.

In order to merge high strength with biological activity in a single implant, a current strategy envisages the development of zirconia-based scaffolds, to provide the necessary mechanical properties, coated by a calcium phosphate thin layer, to provide bioactivity and osteointegration. Several examples have already been reported in the literature [11,12,13,14,15,16,17,18,19], and some of these scaffolds have been already submitted to in vitro and in vivo tests [11,13,14,18,19], fully supporting the effectiveness of this approach. However, some studies indicated the occurrence of an undesired reaction between hydroxyapatite and zirconia components, providing second phases (such as α/β-tricalcium phosphate and calcium zirconate) which reduce the mechanical strength and promote biodegradability [17]. Therefore, some authors performed a double coating process, by interposing a fluorapatite layer between the zirconia and hydroxyapatite ones, able to suppress the mentioned undesired reaction [17,20,21].

The development of macro-porous scaffolds benefits more and more from the new 3D printing technologies, thanks in particular to the possibility to customize patient needs, to perfectly control the inner architecture of the scaffolds, and to have almost no limits in terms of geometric complexity [22,23]. Some previous studies already demonstrated the feasibility of preparing zirconia scaffolds via different 3D printing techniques, which were then coated with bioactive ceramic layers. In particular, Kocyło et al. [17] fabricated zirconia lattice samples via direct ink writing, which were subsequently submitted to the previously mentioned double coating process, with fluorapatite and then hydroxyapatite. Similarly, Gaddam et al. [24] fabricated lattice zirconia scaffolds by robocasting, which were further coated by a bioactive glass layer. Zhang et al. [25] used DLP to fabricate zirconia simple structures with rounded pores of 1 mm in size. Sakthiabirami et al. [26] fabricated lattice zirconia scaffolds by a hot-melt air-extrusion 3D printer, and then coated the porous structures with a zinc-doped HA/glass composite layer.

Among all the 3D printing methods, vat-photopolymerization techniques, such as stereolithography (SL) and digital light processing (DLP) are known to be the most advanced in the fabrication of engineering ceramics, due to high precision and accuracy in the printed parts [27]. While SL printers generally trace out a path with the laser to cure the designed geometry, DLP cures an entire layer at once, making it more rapid.

Therefore, in this work, DLP was exploited to fabricate zirconia scaffolds. In relation to previous literature, and fully exploiting the potential of this technique, the novelty of the work lies in the direct comparison between simple lattice scaffolds and complex trabecular structures. Furthermore, a one-step process was used to coat zirconia scaffolds with a hydroxyapatite layer, after coating optimization, by controlling the calcination temperature, thus avoiding surface treatments and unnecessary intermediate steps and layers previously investigated in the literature. Hence, a comparison between different scaffold geometries (i.e., lattice and trabecular) in terms of mechanical and physical properties, and coating morphology is here discussed.

## 2. Materials and Methods

### 2.1. Materials

A 3 mol% yttria-stabilized zirconia powder (CY3Z, Saint-Gobain ZirPro, Le Pontet, France) was used as starting material. The slurries were prepared by mixing suitable amounts of CY3Z with a commercial photocurable resin (ADMATEC Europe BV, Nobelstraat, The Netherlands). A commercial dispersant (Disperbyk-103, BYK Chemie, Wesel, Germany) was used as well to obtain a satisfactory viscosity and high solid loading.

Slurries were prepared according to the procedure reported in a previous work: first, blank resin and dispersant were mixed with a mechanical stirrer, then, the powder was slowly added to the liquid mixture and, finally, the ceramic slurry was planetary milled for three hours at 350 rpm [28]. Slurry solid loading was fixed at 40.5 vol% using 1 wt% of dispersant, respect to the zirconia powder.

Coatings were prepared using a commercial hydroxyapatite (Ca_9.60_(HPO_4_)_0.40_(PO_4_)_5.60_(OH)_1.60_, HA) powder (CAPTAL-S, Plasma Biotal, Buxton, UK) having a mean particle size (d_50_) of 2.48 µm as determined via laser granulometry. The HA powder is a calcium deficient HA with a Ca/P atomic ratio of 1.55 [29]. Polyethylene glycol (PEG) (PEG-4000, Sigma–Aldrich, Saint Louis, MO, USA) was used as the binder and ethanol as the solvent.

### 2.2. Scaffolds 3D-Printing, Debinding and Sintering

Scaffolds were designed as simple lattice scaffolds as well as trabecular structures, these last to demonstrate the potential of DLP technology to shape ceramics into complex geometries.

Three types of lattice samples were designed using computer-aided design (CAD) (AutoCAD software, Autodesk, San Rafael, CA, USA). In particular, these lattice samples were characterized by different distances between struts (i.e., 750, 900 and 1050 µm, respectively) but the same strut thickness (i.e., 500 µm), as show in Figure 1. These scaffolds are named according to the strut distance (e.g., L-750 is the lattice scaffold with strut distance 750 µm). Trabecular scaffold design is depicted in Figure 2: this structure well mimics human bones in terms of porosity and structural anisotropy.

Specimens were printed using a DLP-based stereolithographic device (ADMAFLEX 130, ADMATEC Europe BV, Nobelstraat, The Netherlands). In brief, a photocurable ceramic slurry, contained in a reservoir, is spread on a moving tape with a fixed thickness thanks to a doctor blade (in this study a slurry thickness of 125 µm was employed). Each layer of the part under printing is cured using a UV-projector (operating at a wavelength of 405 nm) and is attached to a building plate that moves in the z-direction. Printing parameters were optimized after several preliminary trials, allowing to fix layer thickness, exposure time and LED power at 30 µm, 1000 ms and 250 ‰, respectively. Before sintering, samples underwent a water debinding step, at 40 °C for 24 h, to remove unpolymerized organic fraction, followed by a thermal debinding step up to 1000 °C for the burn-out of the resin [28,30]. Then, all the scaffolds were sintered for 1 h at 1400 °C.

### 2.3. Coating Preparation

After sintering, zirconia scaffolds were coated through immersion into a HA/PEG/ethanol suspension. In particular, alcoholic suspensions at three different HA amounts (10, 20 and 35 wt%, respectively) were prepared, in order to define the best coating conditions. The same PEG:ethanol ratio (1:100) was used in the tree formulations. PEG was dissolved in ethanol and then HA was gradually added until homogeneous dispersion, and kept under continuous stirring for 1 day.

Sintered zirconia scaffolds were immersed in the HA suspension for 2 min under vacuum and dried at room temperature for 3 h to allow solvent evaporation. Then, the coated scaffolds were dried for 1 day at 45 °C before calcination. The effectiveness of the coating procedure can be appreciated in Figure 2, showing the sintered zirconia scaffold (Figure 2b) and the coated zirconia one (Figure 2c), the last showing a homogeneous bluish color due to the HA particles coating. After drying, the coated zirconia scaffolds were calcined at 1300 °C for 90 min, to allow the coating to well adhere to the substrate. A slow heating cycle (Figure 3) was used to allow a slow PEG decomposition, thus avoiding the coating damage.

### 2.4. Characterizations

The fired density of uncoated zirconia scaffolds was determined via buoyancy method, following the Archimedes principle (Density Determination Kit, Sartorius YDK01, Göttingen, Germany), considering a theoretical density (TD) for tetragonal zirconia of 6.05 g/cm^3^. Geometrical density was calculated dividing samples mass for their volume, measured using a high-precision caliper.

X-Ray diffraction (XRD) analysis was performed both on the raw powders and sintered samples using a Pan’Analytical X’Pert Pro instrument (Pan’Analytical, Malvern, UK) with CuKα radiation (0.154056 nm) in the 2θ range 5–70°.

Samples compressive strength was determined using an electromechanical testing system (Zwick Roell 2014, Ulm, Germany) equipped with a load cell of 50 kN. Five sintered zirconia scaffolds of each typology were loaded with a crosshead speed of 0.1 mm/min.

In vitro bioactivity tests were performed according to ISO 23317:2014 “Implants for surgery—In vitro evaluation for the apatite-forming ability of implant materials” [31]. The simulated body fluid (SBF) solution was prepared according to the standard, in order to obtain an ion concentration similar to that of human blood plasma. Lattice and trabecular specimens were placed in transparent plastic bottles and soaked in the SBF solution, at 37 °C, in static conditions and observed after different time intervals (1 week, 2 weeks, 4 weeks, respectively). The required volume of the SBF solution was calculated as a function of the apparent surface area of the specimen, considering 1 mm^3^ of SBF for 0.01 mm^2^ of apparent surface. After soaking, the samples were washed under a gentle flow of ultrapure deionised water to remove the residual SBF ions and were dried at ambient temperature subsequently.

Scaffolds and coatings microstructure, as well as bioactivity properties were investigated by means of a FE-SEM (FE-SEM Hitachi S4000, Tokyo, Japan) after Pt sputtering.

## 3. Results and Discussion

### 3.1. Scaffolds Physical-Mechanical Properties

Both lattice and trabecular scaffolds were successfully printed and sintered, as shown in Figure 1 and Figure 2, respectively. DLP consists in the printing of consecutive layers to obtain the desired geometry. For example, an FE-SEM micrograph of an L-750 scaffold is reported in Figure 4a. At higher magnifications (inset of Figure 4a), the consecutive layers composing the scaffold are still recognizable, but their uniformity is evident as well.

Lattice scaffolds were designed considering three different struts distance and, consequently, different porosities (Figure 1 and Table 1). In particular, L-750, L-900 and L-1050 have a nominal porosity (i.e., as designed per CAD model) of 57, 61 and 64%, respectively. On the other side, trabecular scaffolds are anisotropic structures with a not-constant geometric structure. In this study, trabecular scaffolds with a nominal porosity of 76% were employed (Figure 2 and Table 1). After sintering 1 h at 1400 °C, all the lattice scaffolds reported a quite high Archimedes density ranging between 96–97%TD (Table 1). In a previous work we demonstrated that full densification of the same zirconia formulation occurred by sintering at 1550 °C for 1 h [28]. However, in the present study, the sintering temperature was intentionally lower than the maximum densification one, in order to provide some residual pores on the scaffolds surface, to promote the coating adhesion, thus avoiding any physical and/or chemical, modification of the surface. Therefore, nanometric and submicrometric pores can be easily detected in the scaffold microstructure, as shown in Figure 4b,c. From the higher magnification micrograph (Figure 4c) the submicrometric/nanometric size of the zirconia grains can be easily observed.

Trabecular scaffolds reported a lower Archimedes density (90%TD) compared to lattice scaffolds (Table 1) due to the difficulties in cleaning scaffolds from the uncured slurry in such more complex structure. Indeed, the presence of uncured slurry led to the occlusion of some internal pores resulting in a less dense microstructure compared to the photopolymerized one.

All the scaffolds reported a difference between nominal porosity and geometrical porosity (Table 1), i.e., the actual porosity of sintered samples. This difference is due to a discrepancy between the nominal geometry (i.e., that deriving from the CAD model) and the printed one. In fact, it is well known that due to scattering phenomena and overcuring of the photopolymeric ceramic slurry, the printed parts differ in the order of microns from the designed model [32,33]. Moreover, the shrinkage that occurs during sintering should be taken into account as well. In a previous study [28], the shrinkage after sintering 1 h at 1550 °C was measured to be approx. 23%. From the dilatometric curve, a shrinkage of approx. 18% can be determined at 1400 °C [28]. This value is in line with the difference between nominal porosity and the measured porosity (Table 1) and further confirmed by the image analysis performed on lattice scaffolds after sintering to determine pores size. Indeed, the measured pore sizes were: 610 ± 5 µm; 740 ± 2 µm; 865 ± 4 µm for L-750, L-900 and L-1050, respectively.

Considering mechanical properties, as expected, higher the porosity and lower the compressive strength (Table 1). Indeed, a sharp decrease in compressive strength was determined moving from lattice scaffolds to trabecular ones with an almost linear relationship existing between compressive strength and scaffolds porosity (Figure 5). Lattice scaffolds have high compressive strengths ranging between 130–180 MPa, approximatively (Table 1). Indeed, Kocyło et al. [17] reported a compressive strength ranging from 20.8 to 62.9 MPa, depending upon the porosity level, for zirconia scaffolds prepared via direct ink writing and sintered for 2 h at 1450 °C. On the other side, Zhang et al. [25] prepared zirconia scaffolds via DLP and obtained a compressive strength of approx. 30 MPa, for zirconia scaffolds sintered 2 h at 1400 °C. In their study, scaffolds were printed with a similar pore size opening (i.e., 1000 µm) but with a lower solid content compared to the one investigated in this study (65 wt% and 79 wt%, respectively) and this difference could be responsible for the different mechanical properties. Despite the slight difference in terms of porosity, a decrease in compressive strength of approx. 30% was measured between L-750 and L-1050 samples. Interestingly, some authors also prepared lattice HA [34] or Zirconia/HA [35] composite scaffolds via DLP obtaining lower compressive strengths. In particular, Feng et al. [34] achieved compressive strengths ranging between 14–21 MPa depending on the HA solid content for scaffolds sintered at 1300 °C. For Zirconia/HA composites, Cao et al. [35] measured a compressive strength decrease at increasing HA content (from 50 to 15 MPa, from 10 to 30 wt% of HA) while neat zirconia lattice scaffolds had a compressive strength of 40 MPa. On the other side, trabecular scaffolds exhibited the lowest mechanical strength among the investigated structures. This mechanical behaviour can be attributed both to the higher porosity, compared to lattice scaffolds, and to the different geometry, which was anisotropic with mainly very thin walls (approx. between 150 and 300 µm). However, mechanical strength values are in the range of those of human bones that are reported to be 100–200 MPa, for cortical bone, and 2–20 MPa, for cancellous bone [36]. Further, these results highlight the potential of the DLP technique in fabricating scaffolds with the desired physical and mechanical characteristics, by simply modulating the starting CAD file, to control porosity amount, size and struts thickness.

### 3.2. Coating Optimization

As detailed in the Experimental section, the printed scaffolds were coated with a HA layer, by investing three HA solid loadings in the alcoholic suspensions: 10, 20 and 35%, respectively. While a uniform layer was correctly deposited on the zirconia substrate in all three cases, FE-SEM micrographs (Figure 6) highlight some differences by the microstructural point of view. In particular, 10% HA coating results in a porous and discontinuous hydroxyapatite layer (Figure 6a). On the contrary, the 35% HA coating provides a less porous coating, with more connections between HA grains (Figure 6c). However, several cracks are visible in this samples (Figure 6d, white arrows), due to the HA shrinkage during sintering. In fact, during densification, the shrinkage of the coating layer was constrained by the already densified zirconia substrate, leading to tensile stress and consequently to cracking. A similar behavior was observed for HA coatings on Titanium alloys implants [37,38]. Therefore, after this preliminary optimization, the 20% HA coating (Figure 6b) was chosen as the optimal HA concentration for the coating of zirconia scaffolds, providing a highly porous microstructures within a continuous HA layer.

FE-SEM micrographs of samples cross sections are depicted in Figure 7, which were performed to assess the interface between HA coating and the zirconia substrate. HA grains are well attached onto zirconia ones and appear to be strongly adhered to the substrate without cracks or delamination at the interface. Indeed, at higher magnifications (Figure 7b), the good interaction between HA grains, that are in the order of microns, and zirconia nanometric grains is evident. Moreover, still from Figure 7, the high interconnected porosity of the coating can be easily recognized, with such porosity playing a pivotal role in the coating bioactivity.

To evaluate the effects of the sintering temperature on the HA powder crystalline phases, XRD patterns of the as-received and calcined (1 h at 1300 °C) HA powders are reported in Figure 8a. Considering the as-received powder, it can be seen that the XRD pattern consists of narrow peaks completely matching the reference pattern of pure HA (JCPDS file #09-0432) and no other phases were detected. After calcination, the powder was predominantly (approx. 59%, as determined in a previous study [29]) composed by β-tricalcium phosphate (β-TCP, JCPDS file #09-0169), with a still significant amount of HA phase and with some traces of α-tricalcium phosphate (α-TCP, JCPDS file #09-0348). The formation of these secondary phases was expected, being the starting powder a calcium deficient hydroxyapatite, as previously mentioned (§ 2.1). It is worth mentioning that HA/TCP mixture, known as biphasic calcium phosphate (BCP) is regarded as the optimal composition in the calcium phosphate compounds, as it merges the good mechanical properties of HA with the higher biological properties of TCP, especially the osteoinduction [39] and the bioresorbability [40]. In Figure 8b the XRD pattern, shown in the 15–70 2θ (°) range, of a coated zirconia sample after calcination at 1300 °C, is depicted. Besides the main patterns of the tetragonal ZrO_2_ phase (JCPDS file #88-1007), all other signals were indexed according to the β-TCP phase, showing an even higher decomposition of the non-stoichiometric HA into tricalcium phosphate phase when deposited on a zirconia substrate. As these scaffolds are designed with a zirconia structure as the load-bearing part, the transformation of HA into most biologically active phases is here considered a plus. In addition, no traces of calcium zirconate phase were detected, suggesting that the undesired reaction between the zirconia substrate and the calcium apatite layer did not occur. The formation of this phase depends on the thermal treatment: Kim et al. [21] showed the formation of CaZrO_3_ phase on HA-coated zirconia substrates starting from 1200°C, while Kocyło et al. [17] showed the same reaction occurring at 1350 °C/1 h. In our system, the calcination treatment performed at 1300 °C/1 h was demonstrated to be optimal, since it allowed a good adhesion of the layer without the occurrence of the CaZrO_3_ phase.

### 3.3. SBF Immersion Test

The scaffolds bioactivity was investigated through their immersion in SBF for different time intervals and, subsequently, observing the surface via FE-SEM. All the investigated samples after 7 days of immersion, independently from their geometry, showed apatite nucleation seeds (Figure 9a–c). In particular, apatite nucleation seeds are mainly deposited separately, but some clusters can be recognized as well (white arrows in Figure 9c). Nucleation seeds range between 50 and 100 nm, approximatively (Figure 9b). For comparison, a reference zirconia sample without coating was immersed in SBF as well and no apatite formation was detected, as also reported elsewhere [19].

After 14 days of immersion (Figure 9d–f), spherical crystallites can be recognized on both scaffold geometries (i.e., lattice and trabecular) attesting scaffolds bioactivity. In particular, isolated crystallites keep their spherical geometry while the coalescence of crystallites in their proximity results in a modification from such a spherical morphology. At higher magnifications (Figure 9e,f), the flowerlike morphology can be recognised. After 28 days of immersion (Figure 9g–i), a different density of the apatite covering layer is recognizable, meaning that further crystallization occurs due to a different stage of crystallites growth. Indeed, original spherical crystallites are no longer recognizable, and the density of the plate-like crystals is higher (Figure 9i) compared to samples immersed for 14 days (Figure 9f).

The influence of scaffolds geometry on bioactivity properties was evaluated as well. On both lattice and trabecular scaffolds, the formation of an apatite layer was evident (Figure 10) but some differences can be recognized as well. In particular, in the case of lattice scaffolds (Figure 10a–c) apatite was formed in a uniform and continuous way inside the struts of the scaffold but to a lesser extent on the lateral surfaces. On the contrary, in the case of trabecular scaffold (Figure 10d–f) the apatite layer was formed on all the surfaces. In this case, the presence of convex surfaces and a more complex geometry led to a higher degree of covering. As a possible explanation for lattice scaffolds, since the test was carried out in SBF static conditions, ions could only be driven to move by ion concentration gradient. Therefore, Ca^2+^ and HPO_4_^2−^ ions released from the coating cannot be easily dispersed, resulting in a local relatively higher concentration inside lattice pores compared to the surface.

## 4. Conclusions

In this work, the feasibility of fabricating multi-functional scaffolds, characterized by both high-mechanical properties and high bioactivity is fully demonstrated.

Zirconia scaffolds were fabricated by digital light processing (DLP): besides simple lattice shapes (at different infill amount), trabecular structures were fabricated as well in order to fully exploit the potential of DLP to shape ceramics into complex geometry and to better mimic the bone structure. Samples were sintered at 1400 °C/1 h and subjected to compressive strength tests, which decreased by increasing the scaffolds porosity, and provided values covering those of both cortical and cancellous bone. Thus, the advantage of DLP in the fabrication of tunable structures on the grounds of their desired properties is well demonstrated.

Scaffolds sintered at 1400 °C were characterized by a tiny residual porosity to promote the subsequent adhesion of the coating. Scaffolds were in fact coated by a calcium-deficient hydroxyapatite (HA) layer and submitted to a calcination treatment at 1300 °C/1 h, during which the adhesion of the layer was accomplished, besides the decomposition of HA into β-TCP, characterized by more pronounced biological behavior. Therefore, the final structure successfully joined high compressive strength with high bioactivity, as demonstrated by immersion tests in simulated body fluid.

## Figures and Tables

**Figure 1 jfb-13-00148-f001:**
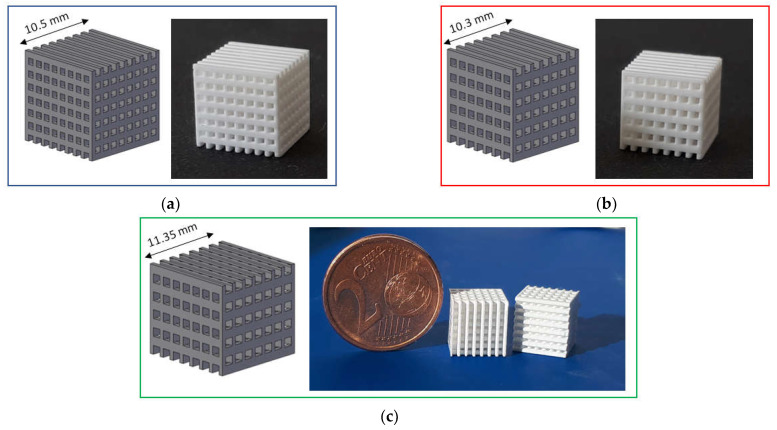
CAD model and printed lattice samples after sintering 1 h at 1400 °C: (**a**) L-750; (**b**) L-900 and (**c**) L-1050.

**Figure 2 jfb-13-00148-f002:**
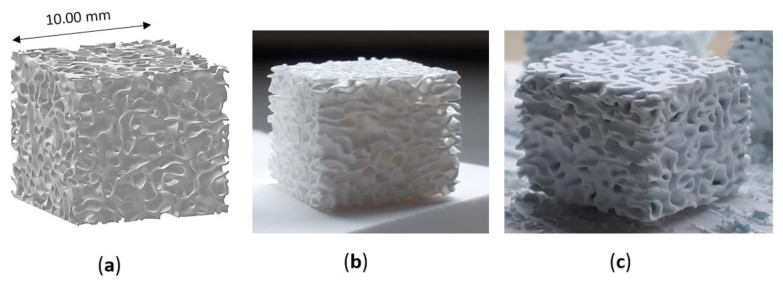
Trabecular scaffold: (**a**) CAD model; (**b**) sintered sample (1 h at 1400 °C) and (**c**) scaffold after coating.

**Figure 3 jfb-13-00148-f003:**
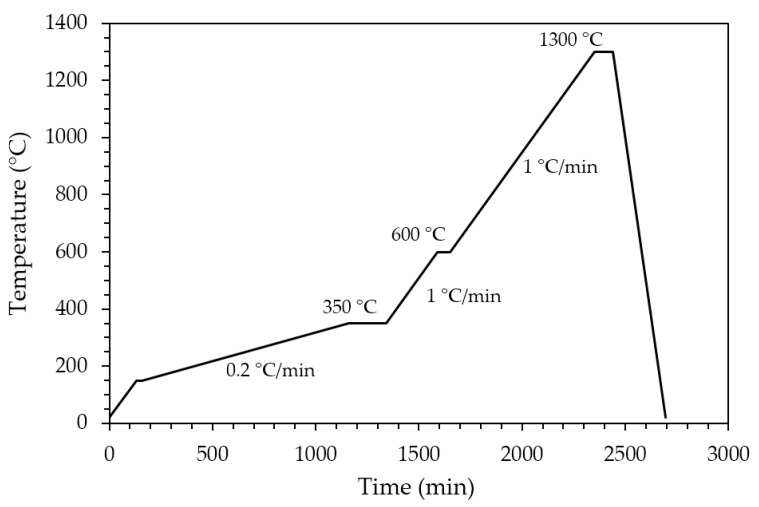
Thermal cycle for PEG decomposition and HA coating sintering.

**Figure 4 jfb-13-00148-f004:**
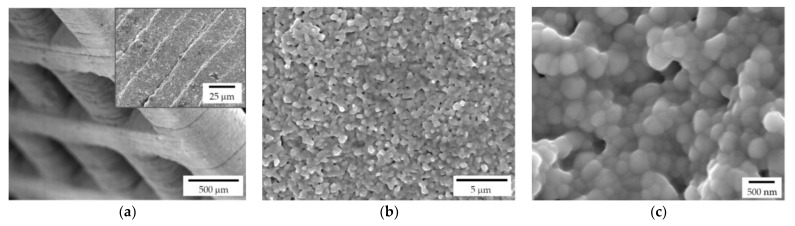
FE-SEM micrographs of a sintered zirconia scaffold (L-750): (**a**) low magnification picture showing the lattice structure and consecutive layers deriving from the 3D printing process (inset); zirconia microstructure at lower (**b**) and higher magnification (**c**), to highlight the nanometric pore size and the tiny residual porosity.

**Figure 5 jfb-13-00148-f005:**
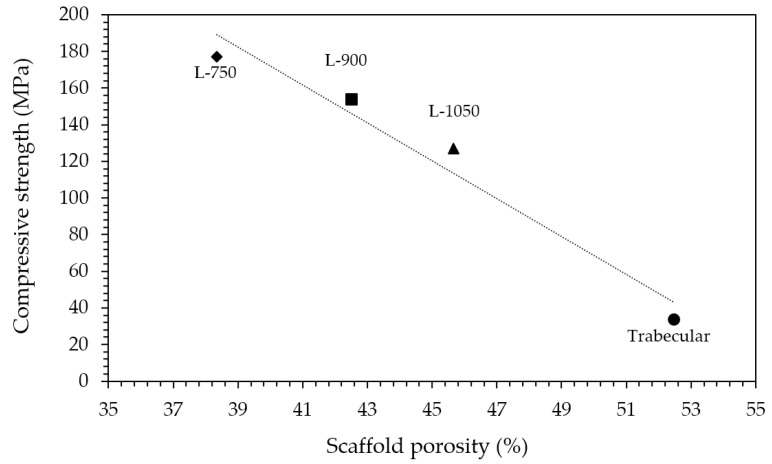
Relationship between compressive strength and scaffolds porosity.

**Figure 6 jfb-13-00148-f006:**
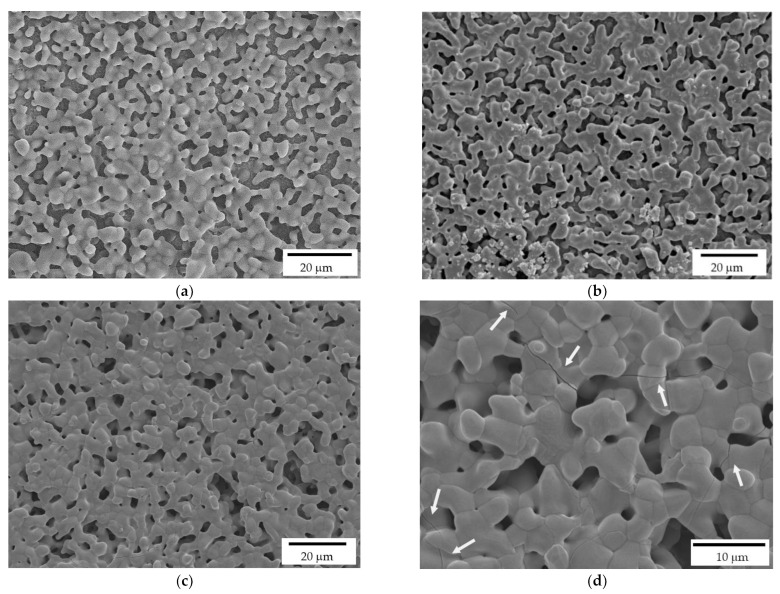
FE-SEM micrographs of HA coatings, deposited on the zirconia scaffolds, and prepared at 10% (**a**), 20% (**b**) and 35% (**c**,**d**) HA solid loading in the ethanol suspensions. White arrows indicate the cracks on the coating produced by the more charged suspension.

**Figure 7 jfb-13-00148-f007:**
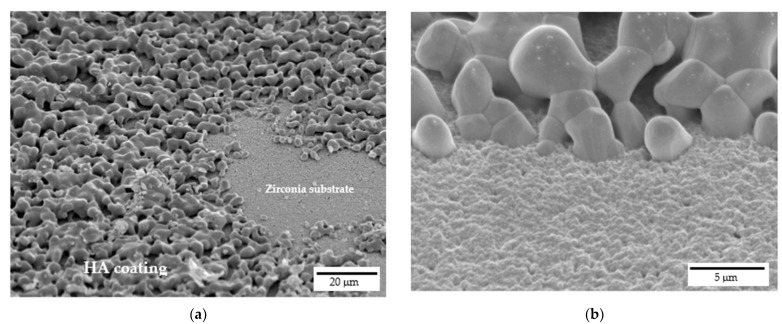
Interface between 20% HA coating and zirconia scaffold: (**a**) cross-sectional view at lower magnifications and (**b**) detail at higher magnifications.

**Figure 8 jfb-13-00148-f008:**
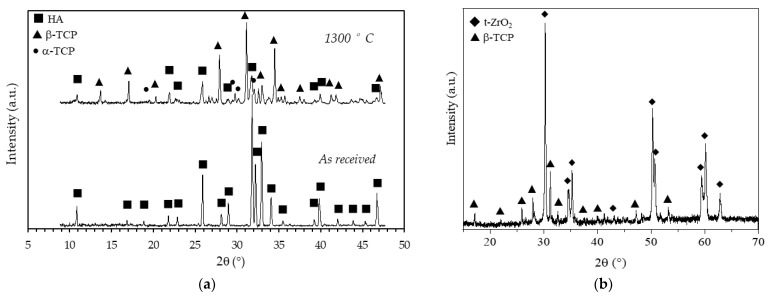
X-Ray diffraction patterns of (**a**) as received and calcined (90 min at 1300 °C) HA powder and (**b**) of a coated zirconia sample after calcination at 1300 °C.

**Figure 9 jfb-13-00148-f009:**
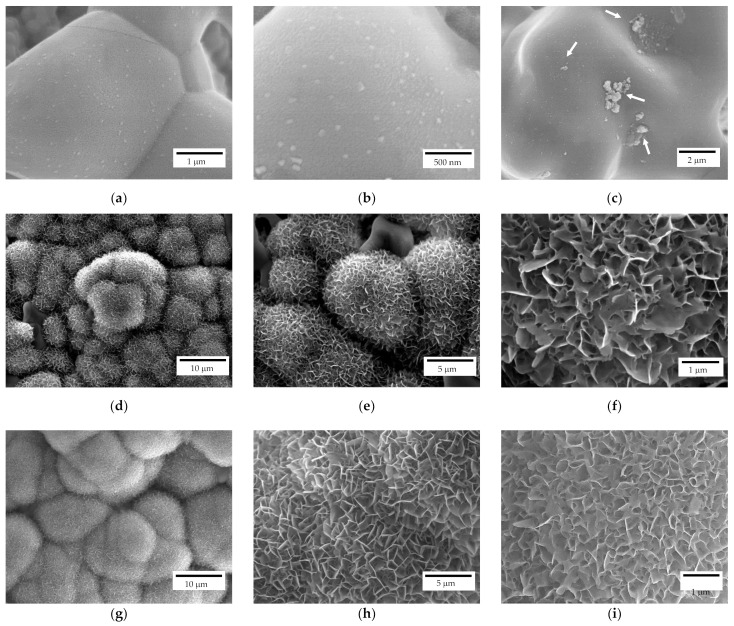
FE-SEM representative micrographs of a scaffold after 7 (**a**–**c**), 14 (**d**–**f**) and 28 (**g**–**i**) days of immersion in SBF.

**Figure 10 jfb-13-00148-f010:**
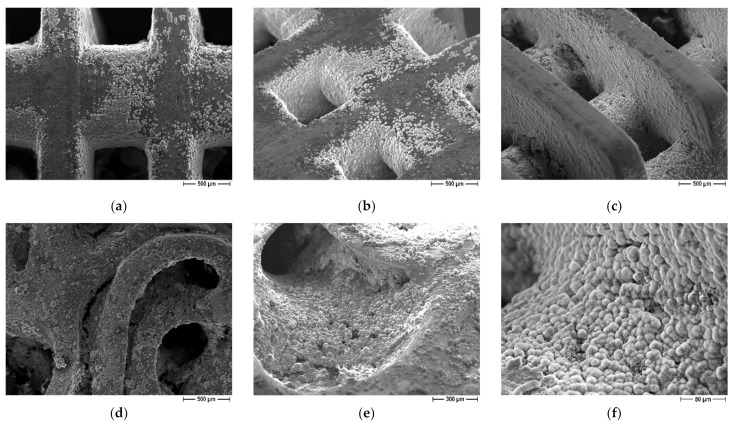
FE-SEM micrographs of coated scaffolds after immersion in SBF, influence of scaffold geometry: (**a**–**c**) lattice scaffolds (**d**–**f**) trabecular scaffold.

**Table 1 jfb-13-00148-t001:** Scaffolds physical and mechanical properties.

Sample	Struts Distance (µm) *	Nominal Porosity (%) *	Archimedes’ Density (g/cm^3^) (%TD) °	Geometrical Density (g/cm^3^) °	Porosity (%) °	σ_c_ (MPa) °
L-750	750	57	5.89 ± 0.06 (97)	3.63 ± 0.06	38	177 ± 31
L-900	900	61	5.85 ± 0.01 (97)	3.36 ± 0.05	43	154 ± 28
L-1050	1050	64	5.79 ± 0.03 (96)	3.14 ± 0.02	46	128 ± 12
Trabecular	-	76	5.47 ± 0.05 (90)	2.60 ± 0.08	52	34 ± 8

* As per CAD model. ° After sintering 1 h at 1400 °C. TD = 6.05 g/cm^3^.

## Data Availability

Not applicable.

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
