# Peer review of "DLP Fabrication of Zirconia Scaffolds Coated with HA/β-TCP Layer: Role of Scaffold Architecture on Mechanical and Biological Properties"

_jfb, 2022, doi:10.3390/jfb13030148_

Round 1

Reviewer 1 Report

I am happy to write to you. In connection with a manuscript entitled; " DLP fabrication of zirconia scaffolds coated with HA/β-TCP 2 layer. Role of scaffold architecture on mechanical and biological properties”, the authors did not make the necessary corrections as well as did not answer pressing questions regarding the preparation, and accordingly the major reviews are outlined which can be summarized as follows:

1.     The title is not suitable. I suggest to be "Study of Microstructural and Electrical Properties of Silver Substituted Hydroxyapatite for Drug  Delivery Applications.

2.     There is a mistake in the title, it must be corrected “DLP fabrication of zirconia scaffolds coated with HA/β-TCP 2 layer. Role of scaffold architecture on mechanical and biological properties

3.     Abstract must consist of the purpose and methodology along with significant findings, which is lacking in the current article. need to be changed.

4.     The introduction is too small and does not reflect the importance of hydroxyapatite as a valuable biomaterial. It should therefore be extended by adding a long paragraph discussing what is meant by hydroxyapatite, its importance and general applications. The following references can help you:

·        D.E. Abulyazied, A. Alturki, R.A. Youness, H. Abomostafa, J. Inorg. Organomet. Polym. Mater. 31 (2021): 4077-4092.

·        M.A. Taha, R.A. Youness, M. Ibrahim, Ceram. Int. 46 (2020): 23599-23610.

·        

5.     The authors did not mention the novelty of their work. Thus, the novelty of this work should be clearly discussed.

6.     The language must be carefully reviewed.

Reviewer 2 Report

Some corrections are required;

1. There are English mistakes throughout the manuscript. Please check carefully.

2. Schematically show the Digital Light Processing (DLP) process.

3. Some abbreviations are not explained; L-750, L-900, L-1050, PEG etc. Please check and explain each and every abbreviation used in this manuscript.

4. Although porosity as given in Table 1 is from 38-52% however the strength is quite high as compared to previous studies. Please check carefully and show the stress-strain curves as well. 

5. Porosity mentioned in Table 1 is from 38-52% however the density is from 90-97%. Correct or explain it.

6. For Fig. 6, please also show the FESEM micrograph without a coating sample as well.

7.  I found some other papers with similar work as well; Additive manufacturing of hydroxyapatite bioceramic scaffolds: Dispersion, digital light processing, sintering, mechanical properties, and biocompatibility, Fabrication and properties of zirconia/hydroxyapatite composite scaffold based on digital light processing. Please compare your work with already published work that has similarities with your work.

8. For the author's contributions, write full names and detail of contributions. The same name should not be repeated in this section.

9. Papers of some researchers in this field e.g. Prof. Aldo Buchachini, Dr. Atiq-ur-Rehman should also be cited.

10. Reference style is not uniform, please check and correct it.

Reviewer 3 Report

It is an interesting work, with nice results, but some details are not clear presented:

 1. I suggest for the authors to summarise in one table al the prepared and analysed samples, indicating their relevant particularities

2. what about the shrinkage of the sintered zirconia samples? How the sintering of samples influences the sizes of the zirconia samples?

3. It is known, as the authors described, the Ca-deficient samples have less thermal stability as the the normal one (with Ca/P=1,67). The decomposition of Ca-deficient samples is not predictable, regarding the ratio of TCP and HA. What is the proposal of the authors regarding the reproducibility of the samples?

4. All the samples tested in SBF were coated with HA-slurry? What about the uncoated samples? Were they also immersed in SBF solution?

Reviewer 4 Report

This article deals with the structure of the zirconia scaffolds coated with calcium phosphates and their biological and mechanical properties.

Although the objectives of this work seem to be interesting and the research is appropriate, several parts of the manuscript should be improved and rewritten.

Therefore, I cannot recommend this paper to be published as it is in Journal of Functional Biomaterials.

Specific comments:

  1. Abstract and title

Abstract text vs title is quite confusing. Information about the real composition of the CaPs coating is not mentioned in the abstract.

2.    Introduction

The introduction is well written. However, the Authors don’t show the original input of their work. I also strongly recommend emphasizing the novelty of their work.

  1. Experimental Section

Why the Authors decided to sinter the scaffolds at 1300°. It is known that HA decomposes above 1200°. Please, clarify.

4.     Results and Discussion

The percentage composition of the CaPs layer should be estimated on the basis of the results of PXRD studies. The "biphasic CaP” term does not apply in this case, because according to the diffraction patterns there are at least three crystalline phases. The work lacks broader biological research, in particular on cells (osteoblasts).

Round 2

Reviewer 1 Report

Greetings

The authors did not make the necessary adjustments carefully. So I recommend rejecting the manuscript.

Reviewer 2 Report

I feel the authors did not give full consideration to my last comments. They did not schematically show the DLP process, lines 166-170 of the revised manuscript are not related to DLP. Reply to comments 7-8 are missing in the reply to reviewers comments file, papers of Prof. Aldo and Atiq as I mentioned last time, are neither cited nor discussed in the text. I asked authors not to write one name for more than one time in the author's contribution but no changes are made. I would suggest authors to make these changes.

Reviewer 3 Report

The authors answered the questions and corrected the manuscript as well.

Author Response

Authors thank Referee 3 for the time dedicated to review the manuscript and for her/his final approval

Reviewer 4 Report

The Authors have corrected the manuscript according to the suggestions.

Author Response

Authors thank Referee 4 for the time dedicated to review the manuscript and for her/his final approval

Round 3

Reviewer 1 Report

Greetings

The authors carefully made all necessary adjustments. Therefore, I recommend that you accept the manuscript.

Reviewer 2 Report

I am satisfied with the authors' reply to my comments.